# Antibiotic Resistance Profile of RT 027/176 Versus Other *Clostridioides difficile* Isolates in Silesia, Southern Poland

**DOI:** 10.3390/pathogens11080949

**Published:** 2022-08-22

**Authors:** Małgorzata Aptekorz, Krzysztof Sacha, Zygmunt Gofron, Monika Kabała, Celine Harmanus, Ed Kuijper, Gayane Martirosian

**Affiliations:** 1Department of Medical Microbiology, School of Medicine in Katowice, Medical University of Silesia, 18 Medyków str., 40-752 Katowice, Poland; 2Department of Medical Microbiology, Leiden University Medical Center, 2333 ZA Leiden, The Netherlands

**Keywords:** *Clostridioides difficile*, hypervirulent ribotypes, antibiotics, multidrug resistance

## Abstract

*Clostridioides difficile* is an important health care-associated pathogen. The aim of this study was to analyze the antibiotic susceptibility of *C. difficile* isolates from feces of patients from 13 hospitals in Silesia, Poland. The incidence of CDI per 100.000 people in Silesia in 2018–2019 was higher than the average in Poland (39.3–38.7 vs. 30.2–29.5, respectively). The incidence doubled from 26.4 in 2020 to 55.1 in 2021. Two hundred and thirty stool samples tested positive for GDH (glutamate dehydrogenase) and toxins were cultured anaerobically for *C. difficile*. The isolates were characterized, typed, and tested for susceptibility to 11 antibiotics by E-test (EUCAST, 2021). The genes of toxins A/B and binary were detected by mPCR. Of 215 isolates, 166 (77.2%) were classified as RT 027 and 6 (2.8%) as related RT 176. Resistance to ciprofloxacin (96.7%), moxifloxacin (79.1%), imipenem (78.1%), penicillin (67%), and rifampicin (40.5%) was found. The *ermB* gene was detected in 79 (36.7%) strains. Multidrug resistance (MDR) was confirmed in 50 (23.3%) strains of RT 027 (94%). We concluded that a high prevalence of MDR among hypervirulent RT 027/176 *C. difficile* was found in the Silesian region of Poland, emphasizing the need to enhance regional infection control on CDI and antibiotic stewardships.

## 1. Introduction

*Clostridioides difficile* is an important pathogen associated with health care, and is responsible for a wide spectrum of diseases, ranging from mild diarrhea to complications such as pseudomembranous colitis and toxic megacolon [1].

In recent years, more attention has been brought to these infections due to the increase in incidence and mortality among hospitalized patients with CDI (*Clostridioides difficile* infection). The main virulence factors of *C. difficile* include enterotoxin A (TcdA) and cytotoxin B (TcdB). The genes that encode toxins are located in the PaLoc region (pathogenicity locus) of 19.6 kbp genomic DNA [2]. Some strains of *C. difficile* (6–30%) additionally produce a binary toxin (CDT) that possesses two subunits: CDTa (increasing the adherence of *C. difficile*) and CDTb (responsible for the binding and transfer of CDTa into the cytoplasm of the target cells) [3]. The usage of antibiotics—especially fluoroquinolones, third-generation cephalosporins, and clindamycin—is associated with a high risk of CDI development [4,5].

The incidence of CDI per 100.000 people in Silesia in 2018–2019 was higher than the average in Poland (39.3–38.7 vs. 30.2–29.5, respectively). The incidence doubled from 26.4 in 2020 to 55.1 in 2021 (http://wwwold.pzh.gov.pl/oldpage/epimeld/index_p.html accessed on 31 December 2021). It is necessary to take into account the significant increase in incidence of CDI during the COVID-19 pandemic. The widespread antibiotic and disinfectant use, as well as the direct alteration of SARS-CoV-2 on microbiota, constituted a crucial risk factor for CDI [6].

Recently, *C. difficile* was recognized by CDC as one of the top five urgent antibiotic-resistant threats in the USA [7]. Antibiotic resistance of *C. difficile* strains plays an important role in the pathogenesis and spread of CDI. The resulting selection pressure predisposes the emergence and spread of resistant strains. The percentage of MDR (multidrug resistance) in *C. difficile* is noted to be between 2.5 and 66% in various countries [8,9]. A European prospective study of CDI indicated that 55% of resistant clinical isolates had MDR in 2005 [10]. *C. difficile* strains with MDR that were resistant to rifampicin were described in Italy [11] and other countries.

Some epidemiological data suggest that quite a few *C. difficile* ribotypes have been associated with specific antibiotic resistance, including strains resistant to fluoroquinolones (RT 027 and RT 017), rifampicin (RT 027), and clindamycin (RT 017) or tetracycline (RT 078) [12]. The global spread and outbreak of hypervirulent *C. difficile* strains belonging to RT 027 is associated with antibiotic usage, especially with the massive use of fluoroquinolones [9,13]. However, there are also outbreaks of CDI in both American and European healthcare facilities, which were caused by strains belonging to other ribotypes—such as 001, 002, and 014/020, as well as 017, 018, 106, 176, and 244 [2,14,15,16]. Exposure to antimicrobials plays a significant role in the pathogenesis of CDI, and resistance should be considered to be a predisposing factor for CDI development.

The aim of this study was to analyze the susceptibility profile of *C. difficile* isolates from fecal samples of hospitalized patients in hospitals in the Silesian region of Poland.

## 2. Results

### 2.1. C. difficile Strains

After discarding the repeated samples, 215 *C. difficile* isolates taken from fecal samples of hospitalized patients suspected for CDI were involved in this analysis: 120 were from women aged 20 to 92 years (median = 77) and 95 were from men aged 16 to 91 years (median = 71). A history of previous fluoroquinolone treatment was documented in 30/215 patients with positive *C. difficile* cultures. Table 1 depicts the characteristics of the cultured isolates. Among the cultured *C. difficile* strains, 166 (77.2%) were classified as RT 027 and 6 (2.8%) as the related RT 176. The remaining strains belonged to 14 different PCR ribotypes: RT 014 (3.7%; 8/215), RT 023, (2.3%; 5/215), RT 001 (1.4%; 3/215), RT 018 (1.4%; 3/215), RT 282 (1.4%; 3/215), RT 005 (0.9%; 2/215), RT 010 (0.9%; 2/215), RT 052 (0.9%; 2/215), RT 002 (0.5%; 1/215), RT 015 (0.5%; 1/215), RT 045 (0.5%; 1/215), RT 046 (0.5%; 1/215), RT 076 (0.5%; 1/215), and RT 081 (0.5%; 1/215). For nine (4.2%) of the *C. difficile* strains, the ribotype could not be determined. The analysis confirmed the simultaneous presence of genes encoding A, B, and binary toxins in 83.7% (180/215) of the examined *C. difficile* strains.

### 2.2. Detection of ermB Gene

Gene *ermB* was detected in 79/215 strains. In 75 of the strains, the genes encoding toxins A, B, and binary toxin were also detected. In one strain (RT 010), all toxin genes were absent. In three of the remaining strains (RT 001, RT 014, and RT 046), only the toxin A and B genes were detected. The remaining 136 *C. difficile* strains were *ermB*-negative (Table 1).

### 2.3. Antibiotic Susceptibility Testing

The susceptibility of *C. difficile* strains to 11 antibiotics was determined. All strains were sensitive to metronidazole, vancomycin, amoxicillin/clavulanic acid, and piperacillin/tazobactam. The geometric means (GMs) of metronidazole for all the tested ribotypes were 0.68 μg/mL (RT 027—1.0 μg/mL; RT 176—1.1 μg/mL; other toxigenic strains—0.13 μg/mL; nontoxigenic strains—0.33 μg/mL). The GM of vancomycin for all the tested ribotypes was 0.25 μg/mL (RT 027—0.26 μg/mL; RT 176—0.16 μg/mL; other toxigenic strains—0.22 μg/mL; nontoxigenic strains—0.36 μg/mL) (Table 2).

One hundred and seventy-four strains were resistant to erythromycin and 127 to clindamycin. The GMs of erythromycin and clindamycin for all the studied strains were 76.19 μg/mL and 19.07 μg/mL, respectively. In the *ermB-*positive strains, the GMs were higher compared with the *ermB-*negatives: 150.15 vs. 51.38 μg/mL for erythromycin and 71.46 vs. 8.85 μg/mL for clindamycin, respectively. Two hundred and eight strains were resistant to ciprofloxacin, and 170 to moxifloxacin. The GM values of ciprofloxacin and moxifloxacin were calculated and compared in *ermB-*positive and *ermB*-negative *C. difficile* strains: ciprofloxacin was 30.25 vs. 27.45 μg/mL and moxifloxacin was 22.17 vs. 10.45 μg/mL, respectively. Of the strains tested, 168 were resistant to imipenem and 144 to penicillin G. Among the studied strains of *C. difficile*, 40.5% (87/215) showed resistance to rifampicin and the majority (almost 94%) belonged to RT 027 (Table 2). MDR—coresistance to moxifloxacin, clindamycin, erythromycin, rifampin, and imipenem—was confirmed in 50 (23.3%) strains (47 strains of RT 027, and 1 each of 176, 010, and 001); among them, in 36 (72%) isolates belonging to RT 027, presence of the *ermB* gene also was confirmed.

## 3. Discussion

Antimicrobials are strong inducers of CDI, reducing the anaerobic microbiota of the intestine (e.g., *Bacteroides* spp. and *Bifidobacterium* spp.) while sparing some facultative anaerobes (e.g., *Enterococcus* spp.). *C. difficile* resistance to antimicrobial agents (such as macrolide–lincosamides–streptogramin B MLS_B_, fluoroquinolones, tetracyclines, chloramphenicol, or beta-lactams) may be the result of the presence of resistant genes which are transmitted by bacterial chromosomes and mobile genetic elements, mutations, and changes on antibiotic targets and/or metabolic pathways of *C. difficile* and in biofilm production [17,18,19]. Horizontal gene transfer has been suggested to play a key role in the spread of antimicrobial resistance (AMR), both within *C. difficile* and among gut bacteria in general [20].

### 3.1. ermB and Resistance to Macrolides and Lincosamides

Three groups of genes encoding MLS resistance have been described. Based on sequence similarity, *erm* genes responsible for cross resistance to MLS_B_ have been divided into separate classes—often associated with a specific type of bacteria. The only exceptions are the *erm* genes belonging to class B, for which their presence has been observed in numerous bacteria, indicating their potential for translocation between different genera [21]. This is confirmed by the research from Imwattana et al. [12] documenting a large variety of *ermB*-positive transposons. In this study, the presence of *ermB*-encoding MLS_B_-resistance was demonstrated in 79 (36.7%) strains, most often (~89%, 70/79) belonging to hyperepidemic RT 027 and closely related to RT 176~6% (5/79). A smaller percentage (28%) of *ermB*-positive strains was confirmed by Spigaglia et al. [10]. However, contrary to our results, most of their strains belonged to RT 001 (35.6%) and RT 012 (11.1%). According to the literature, different ribotypes demonstrated the presence of the *ermB* gene; it seems that clonal expansion of *C. difficile* ribotypes containing *ermB* in some locations was noted—as in the outbreak of RT 001 in the USA, which was described by Gerding et al. [22] in 1999. Polivkova et al. [23], however observed the presence of *ermB* in only 1.8% (2/64) of the RT 176 strains dominant in the Czech Republic.

Among the 79 *ermB-*positive *C. difficile* strains analyzed, in vitro resistance to erythromycin and clindamycin was found in only 62/79 (78.5%) strains belonging to RT 027 (*n* = 60), RT 176 (*n* = 1), and RT 001 (*n* = 1). However, in *ermB*-positive strains, GMs were significantly higher than in *ermB*-negative: 150.15 μg/mL vs. 51.38 μg/mL for erythromycin (*p* = 0.0345) and 71.46 μg/mL vs. 8.85 μg/mL for clindamycin (*p* < 0.0005), respectively.

### 3.2. Resistance to Rifampicin

Resistance to rifampicin is associated with *rpoB* mutations. Researchers suggest that the increasing number of *C. difficile* isolates resistant to moxifloxacin and rifampicin is associated with acquired resistance in vivo [10,24]. The role of rifampicin therapy for diseases other than CDI in the emergence of *C. difficile* resistance to this antibiotic is unclear. However, Obuch-Woszczatyński et al. [25] described 10 cases of CDI at the Specialized Hospital of Lung Diseases and Tuberculosis (SHLDT) in our region. Seven of these patients (all with prolonged rifampicin therapy) were infected with RT 046 strains, known to be highly resistant to rifampicin. An increased number of rifampicin-resistant isolates suggests cautious and moderate use of rifampicin or related rifaximin in the treatment of CDI recurrences [10]. Rifaximin is a nonabsorbable rifamycin antibiotic with excellent activity against *C. difficile*—an alternative to metronidazole and vancomycin. Rifaximin has a potential role in reducing the rate of CDI recurrences, but clinical studies have reported a high resistance rate with a geographical variance in the distribution of rifaximin-resistant *C. difficile* strains [26]. The present study revealed that 40.5% of *C. difficile* strains were resistant to rifampicin, with over 90% of them belonging to RT 027. In our previous study performed in 2016–2017, among the *C. difficile* isolates belonging to RT 027, almost 31.5% were resistant to rifampicin [27]. Unfortunately, no sequence data were available to determine the mechanism of rifampicin resistance and the relatedness of the rifampin resistant isolates. On the other hand, significantly lower percentages of rifampicin-resistant strains were observed in the five-year pan-European long-term surveillance of *C. difficile*, from 13.5% in year one to 10.2–11.8% in observations from years four to five, respectively [28]. The pan-European study showed resistance to rifampicin most often in strains of RT 027, but resistance was also found in isolates belonging to RT 001, 018, 356, 017, 176, and RT 198 [29]. The present study also demonstrated the presence of resistance to rifampicin in more than one-third (33.3%) of RT 176 and 15.4% of other toxin-producing strains (Table 2). Interestingly, resistance to rifampicin in this study was also determined in three out of four toxin-nonproducing *C. difficile* strains. Pan-European data showed an increase in resistance to rifampicin in Hungary (38.7–56.6%), Italy (36.6–47%), the Czech Republic (40–64%), and Poland (37.9% and 44%) from the year 2011 to the year 2014 [29].

### 3.3. Resistance to Beta-Lactams

Various bacterial species have created mechanisms that reduce the effect of beta-lactam antibiotics by the production of enzymes (beta-lactamases) and the modification of PBP proteins (so that the antibiotic would not be able to bind). The resistance of the pathogen to β-lactam antibiotics plays a major role in the development of CDI, but the mechanism of resistance is currently unknown. In the present study, 168/215 (78.14%) of *C. difficile* strains were resistant to imipenem (Table 2). In contrast, Lachowicz et al. [30], showed in 2012 a higher percentage (87.9%) of strains resistant to imipenem; although similarly to our study, the RT 027 dominated. Noteworthy is the observation that in the current analysis, the percentage of imipenem-resistant strains in RT 027 was lower than was demonstrated in the previous study in the Silesian region (82.5% vs. 100%) [27]. However, a nonsignificant but higher percentage of imipenem-resistant strains was described among the RT 176 isolates than other toxin-producing strains (83.3% and 57.5%). All nontoxin-producing *C. difficile* strains were resistant to imipenem. Different results were obtained by Isidro et al. [31] from Portugal; they showed imipenem resistance in only 12.6% of strains, and the majority (22/24) belonged to RT 017. A comparison of resistance to imipenem requires taking into account breakpoints, because EUCAST gave different breakpoints for Gram-positive anaerobes in different years. In this study, we showed a slightly lower percentage (67%) of *C. difficile* strains resistant to another beta-lactam antibiotic, benzylpenicillin. However, these isolates showed in vitro susceptibility to beta-lactam antibiotics with beta-lactamase inhibitors (amoxicillin with clavulanic acid or piperacillin with tazobactam). Similarly, full sensitivity to the combination of piperacillin with tazobactam was also described in 2009 by Roberts et al. [32] from New Zealand. In contrast to our results, they showed 100% resistance to penicillin in the tested strains. Researchers from China also demonstrated good susceptibility to piperacillin/tazobactam [33]. Our study showed good sensitivity to amoxicillin/clavulanic acid and piperacillin/tazobactam in all of the studied *C*. *difficile* isolates.

### 3.4. Metronidazole and Vancomycin Resistance

Of particular concern is the development of resistance to recommended treatment drugs. Mechanisms of metronidazole resistance remain unclear but increasingly appear to be multigene with a role for iron metabolism. Whereas mutations in proteins of peptidoglycan biosynthesis and biofilm formation are responsible for vancomycin resistance [34].

Debast et al. [35], in 2008, among 398 *C. difficile* strains from 22 European countries, described merely six strains from three different countries with metronidazole MIC = 2 μg/mL. Four of the six isolates were characterized as RT 001. The analysis of *C. difficile* susceptibility conducted in 2011–2014 in 22 European countries showed, among almost 3.000 isolates, rarely observed resistance to metronidazole and vancomycin, in six (0.2%) and two (0.1%) cases, respectively, but there was a reduction described in the metronidazole-resistant isolates over the course of the study [29]. However, the latest epidemiological studies conducted in 2016 in 20 European countries have already reported an increased number of resistant strains, including metronidazole resistance (4.6%; 26/569) and one vancomycin-resistant strain [36]. In the present study, isolates that are resistant to metronidazole were not found. However, similar to others [24,28], a higher level of GMs was observed for this antibiotic among RT 027 (1.0 μg/mL; *n* = 166) and also among six RT 176 isolates (1.1 μg/mL). Compared with the GMs of the remaining toxigenic strains (0.13 μg/mL; *n* = 40), these differences were statistically significant (*p* < 0.005). The highest rates of metronidazole resistance in *C. difficile* have been recovered in Asia and North America. Resistance remains rare but varied, with rates up to 40%. It seems to be more common in nontoxigenic strains, with GM values up to nine times higher than in toxigenic strains [34].

Metronidazole resistance in *C. difficile* correlates with the presence of a 7 kb plasmid, pCD-METRO [37], and also depends on the medium used for the susceptibility testing [38].

In our study, the vancomycin GM value for RT 027, RT 176 isolates, and other toxigenic strains were comparable (~0.2 μg/mL) to the results obtained by Freeman et al. [29]. Single cases of reduced susceptibility to vancomycin have been reported in the Czech Republic, Latvia, and Ireland (MIC 4 μg/mL). Instead, the presence of vancomycin-resistant strains and strains with reduced susceptibility (MIC > 8 μg/mL) were recorded in Italy and Spain, mainly among the RT 027, 126, 356, and 001/072 isolates [18]. However, the most common vancomycin resistance was found in North and South America and Asia, which is reflected in the high level of antimicrobial glycopeptide usage in the USA and China [34].

### 3.5. Multidrug Resistance

In the European study performed in 2011, Spigaglia et al. [10] found 26% multidrug resistance (defined as resistance to three or more classes of antimicrobial agents), among 316 isolates belonging to 11 different ribotypes—most often RT 001, RT 017, and RT 012. We obtained similar results—23.3% (50/215) of *C. difficile* isolates were simultaneously resistant to clindamycin, erythromycin, moxifloxacin, rifampicin, and imipenem. In 47/50 cases, MDR was related to RT 027 and one each to RT 176, 001, and 010.

A three-year (2015–2018) study analysis, in 10 Australian microbiology laboratories in five Australian states, showed that the majority of *C. difficile* strains did not exhibit reduced susceptibility to the antimicrobials recommended for CDI treatment (such as vancomycin, metronidazole, and fidaxomicin). In addition, the prevalence of *C. difficile* MDR—defined as resistance to class 3 antibiotics—was low (1.7%; 19/1091) [39]. In the paper published in 2020, a high prevalence of *C. difficile* strains with MDR was described following the extensive use of antimicrobials in hospitalized patients in Kenya [40]. According to Imwattana et al. [12] prevalence of *C. difficile* MDR was highest in clade 4 (C4—61.6%; RT017; 343/557), which was more than three times higher than in clade 2 (RT027; 356/1951)—the clade with the second highest prevalence of MDR (18.3%). Furthermore, *C. difficile* strains from Australia and New Zealand demonstrated the highest antimicrobial resistance compared with strains from Asia, Europe, and the USA (*p* < 0.0001).

## 4. Materials and Methods

In the period from December 2018 to February 2019, stool samples from hospitalized patients suspected of antibiotic-associated diarrhea (AAD) were tested for the presence of GDH antigen. The majority (43%) of AAD patients were hospitalized in the internal medicine wards; 26% in surgery, cardiosurgery, and vascular surgery; and the remaining 31% in the ICU, nephrology, and others. This study was conducted in accordance with Good Clinical/Laboratory Practice Guidelines and the Declaration of Helsinki. The protocols were approved by the Ethical Committee of the Medical University of Silesia in Katowice, Poland (KNW/0022/KB/127/I/12). AAD was defined as ≥ 3 diarrheal stools per 24 h, 2–8 weeks following the previous antibiotic treatment. In the 230 samples positive for the GDH antigen, the presence of toxins A/B of *C. difficile* were determined using two enzyme immunoassays: C. diff. Quick Check Complete (TechLab, Blacksburg, VA 24060, USA—detects GDH at the level of ≥0.8 ng/mL) and C. DIFFICILE TOX A/B II ™ (TechLab, USA). These fecal samples (28.8%) were cultured (after discarding duplicate samples). Fecal samples (~1 ml/or g) were plated on selective *C. difficile* media (CDIF—chromID ™ *C. difficile* and CLO—Columbia agar with cycloserine, cefoxitin, and amphotericin B (bioMérieux, Marcy L’Etoile, France)) and incubated for 48 h at 37 °C under anaerobic conditions (A35 Whitley anaerobic Workstation, UK). Colonies with a characteristic horse odor and yellow-green fluorescence under UV light (microscopically recognized as cylindrical Gram-positive bacilli) were identified in an automatic system—VITEK 2 Compact (bioMérieux, Marcy L’Etoile, France). Isolated *C. difficile* strains were stored for further testing at −80 °C in Microbanks (Microbank ™ Bacterial and Fungal Preservation System, the PRO-LAB DIAGNOSTICS, 3 Bassendale Road Bromborough, Wirral, Merseyside CH62 3QL, UK).

### 4.1. Molecular Examination

DNA were extracted from cultured *C. difficile* bacilli using the QIAamp DNA Mini Kit (Qiagen, Qiagen Str. 1, 40724 Hilden, Germany). To determine the presence of genes encoding *C. difficile* resistance mechanisms to macrolides, lincosamides, and streptogramin B (MLS_B_), a polymerase chain reaction (PCR) was performed using specific primers [22]. In addition, the DNA of *C. difficile* strains were tested by multiplex PCR (mPCR) to detect genes encoding glutamate dehydrogenase, A, B, and binary toxins [41]. The obtained products underwent electrophoretic separation. The results were analyzed in a G: BOX Chemi XR5 gel imaging system (Syngene, Beacon House Nuffield Road Cambridge CB4 1TF, UK). Ribotyping of the tested *C. difficile* was carried out at the Department of Medical Microbiology at the Leiden University Medical Center in the Netherlands. The sequences of the primers for genes encoding resistance mechanisms and toxins used in this study are presented in Table 3.

### 4.2. Statistical Analysis

For statistical analysis, Microsoft Exel calculation suite and TIBCO Software Inc. (2017) (Statistica—data analysis software system, version 13. http://statistica.io accessed on 30 June 2017 were used.

A *p*-value < 0.05 was considered statistically significant.

### 4.3. Antibiotic Susceptibility Determination

The susceptibility of the tested *C. difficile* strains to antimicrobials was determined using a method based on the minimum inhibitory concentration of antibiotics—MIC, performed by the E-test strips (bioMerieux, Marcy L’Etoile, France). The epidemiologic cutoff values, according to EUCAST (European Committee on Antimicrobial Susceptibility Testing, Version 11.0, valid from 1 January 2021), were applied [42]. The study was conducted as recommended by the manufacturer of the E-test strips (https://techlib.biomerieux.com accessed on 29 December 2017), using strips containing metronidazole (range 0.016–256 μg/mL), vancomycin (0.016–256 μg/mL), moxifloxacin (0.002–32 μg/mL), ciprofloxacin (0.002–32 μg/mL), rifampicin (0.002–32 μg/mL), erythromycin (0.016–256 μg/mL), clindamycin (0.015–256 μg/mL), benzylpenicillin (0.016–256 μg/mL), imipenem (0.002–32 μg/mL), amoxicillin with clavulanic acid (0.016–256 μg/mL), and piperacillin with tazobactam (0.016–256 μg/mL). In this study, the following reference strains from the ATCC collection were used: *Clostridium difficile* ATCC 700057, *Bacteroides fragilis* ATCC 25285, and *Bacteroides thetaiotaomicron* ATCC 29741. All control results were within acceptable limits.

## 5. Conclusions

In the Silesian region of southern Poland, the CDI incidence per 100.000 people is higher than the average in Poland (64.2 vs. 55.5, respectively). We found a high prevalence of *C. difficile* strains with MDR that belong to RT 027, with increasing resistance rates to rifampicin. This stresses the need to enhance regional infection control on CDI and antibiotic stewardship in hospitals.

## Figures and Tables

**Table 1 pathogens-11-00949-t001:** Toxin profile (A/B and binary) of *ermB* (+) and *ermB* (−) *C. difficile* strains.

Toxins and *ermB* Genes	Number (%)	PCR RT
A^+^B^+^CDT^+^; *ermB*^+^	75 (34.9)	027^(70)^, 176^(5)^
A^+^B^+^CDT^−^; *ermB*^+^	3 (1.4)	001^(1)^, 014^(1)^, 046^(1)^
A^−^B^−^CDT^−^; *ermB*^+^	1 (0.5)	010^(1)^
A^+^B^+^CDT^+^; *ermB*^−^	105 (48.8)	027^(96)^, 023^(5)^, 045^(1)^, 176^(1)^, X^(2)^
A^+^B^+^CDT^−^; *ermB*^−^	20 (9.3)	001^(2)^, 002^(1)^, 005^(2)^, 014^(7)^, 015^(1)^, 018^(3)^, 052^(2)^, 076^(1)^, 081^(1)^
A^+^B^−^CDT^−^; *ermB*^−^	8 (3.7)	282^(3)^, X^(5)^
A^+^B^−^CDT^+^; *ermB*^−^	1 (0.5)	X^(1)^
A^−^B^−^CDT^−^; *ermB*^−^	2 (0.9)	010^(1)^, X^(1)^
Total	215 (100)	

^(Superscript)^ the number of strains of a given PCR RT; X—the ribotype could not be determined

**Table 2 pathogens-11-00949-t002:** MIC values (μg/mL) and geometric means of antimicrobials against *C. difficile* strains.

*PCR* *Ribotype*	Measure	MIC Results (μg/mL)										
		Metronidazole	Vancomycin	Moxifloxacin	Ciprofloxacin ^1^	Rifampicin	Erythromycin ^1^	Clindamycin ^1^	Benzylpenicillin ^1^	Imipenem ^1^	Amoxicillin/ Clavulanic Acid ^1^	Piperacillin/ Tazobactam ^1^
027 *n* =166 ermB^+^ = 70	Range (μg/mL)	0.016–2	0.016–1	0.094–32	2–32	0.002–32	0.125–256	0.023–256	0.032–3	0.047–32	0.016–6	0.016–16
	GM	1.00 ^a^	0.26	21.62 ^a^	30.33 ^a^	0.21 ^a^	156.77 ^a^	37.84 ^a^	0.78	13.90	0.30 ^a^	2.86
	MIC _50_	1.5	0.38	32	32	0.003	256	256	1.0	32	0.38	4
	MIC _90_	2	0.5	32	32	32	256	256	1.5	32	0.75	6
	No. and % SR	0/0	0/0	150/90.4	164/98.8	79/47.6	154/92.8	116/69.9	117/70.5	137/82.5	0/0	0/0
176 *n* = 6 ermB^+^ = 5	Range (μg/mL)	0.75–1.5	0.047–0.5	0.38–32	32	0.002–32	0.38–256	0.016–256	0.032–2	4–32	0.016–0.75	0.016–6
	GM	1.09 ^b^	0.17	8.58	32	0.05 ^b^	34.31	1.02	0.31	19.21	0.11	0.83
	MIC _50_	1	0.125	32	32	0.002	256	0.125	0.19	32	0.125	0.75
	MIC _90_	1.5	0.38	32	32	32	256	6	1.5	32	0.5	6
	No. and % SR	0/0	0/0	4/66.7	6/100	2/33.3	4/66.7	2/33.3	2/33.3	5/83.3	0/0	0/0
other toxigenic strains *n* = 40 ermB^+^ = 3	Range (μg/mL)	0.016–0.5	0.016–0.75	0.094–32	1.5–32	0.002–32	0.125–256	0.016–256	0.064–3	1–32	0.016–1	0.016–8
	GM	0.13	0.22	2.14	21.24	0.005	3.93	1.85	0.59	9.46	0.19	1.92
	MIC _50_	0.19	0.38	1	32	0.002	0.75	1.5	0.75	32	0.25	3
	MIC _90_	0.38	0.75	32	32	0.003	256	256	1.5	32	0.5	6
	No. and % SR	0/0	0/0	13/32.5	35/87.5	4/10	13/32.5	8/20	23/57.5	23/57.5	0/0	0/0
nontoxigenic strains *n* = 3 ermB^+^ = 1	Range (μg/mL)	0.25–0.38	0.125–1	32	32	0.003–32	256	0.38–256	0.25–4	32	0.25–1.5	3–12
	GM	0.33	0.36	32	32	1.45	256	7.3	1.14	32	0.66	6
	MIC _50_	0.38	0.38	32	32	32	256	4	1.5	32	0.75	6
	MIC _90_	0.38	1	32	32	32	256	256	4	32	1.5	12
	No. and % SR	0/0	0/0	3/100	3/100	2/66.7	3/100	1/33.3	2/66.7	3/100	0/0	0/0
EUCAST (µg/mL) ^2^		>2	>2	>4	>4	>0.004	>8	>4	>0.5	>4	>8	>16

^1^ MICs for Gram-positive anaerobes were used, because for *C. difficile* they are not present in EUCAST; ^2^ resistance according to EUCAST; ^a, b^ Indicates elevated geometric mean MICs relative to other toxin-producing ribotypes; GM—geometric mean; SR—strains resistant.

**Table 3 pathogens-11-00949-t003:** Primers used for PCR in the present study.

Gene	F—Sequence	R—Sequence	Product Size (bp)
**mPCR** [41]			
*gluD*	GTCTTGGATGGTTGATGAGTAC	TTCCTAATTTAGCAGCAGCTTC	**158**
*tcdA*	GCATGATAAGGCAACTTCAGTGGTA	AGTTCCTCCTGCTCCATCAAATG	**629**
*tcdB*	CCAAARTGGAGTGTTACAAACAGGTG	GCATTTCTCCATTCTCAGCAAAGTA GCATTTCTCCGTTTTCAGCAAAGTA	**410**
*cdtA*	GGGAAGCACTATATTAAAGCAGAAGC GGGAAACATTATATTAAAGCAGAAGC	CTGGGTTAGGATTATTTACTGGACCA	**221**
*cdtB*	TTGACCCAAAGTTGATGTCTGATTG	CGGATCTCTTGCTTCAGTCTTTATAG	**262**
**Mechanism MLS_B_** [22]			
*ermB*	AATAAGTAAACAGGTAACGTT	GCTCCTTGGAAGCTGTCAGTA	**688**

## Data Availability

Not applicable.

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
