# Peer review of "Antibiotic Resistance Profile of RT 027/176 Versus Other *Clostridioides difficile* Isolates in Silesia, Southern Poland"

_pathogens, 2022, doi:10.3390/pathogens11080949_

Round 1

Reviewer 1 Report

Clostridioides difficile is a leading nosocomial agent that presents a major threat among healthcare regimens. Currently, the major mode of therapeutic management for Clostridioides difficile infection (CDI) is antibiotic therapy. However, evolution of antibiotic resistance is a serious, plaguing concern for the healthcare workers. In the current manuscript the authors have discussed this antibiotic resistance in C difficile strains Silesia region of Poland. The authors have done a very systematic study to assess the antibiotic susceptibility of these strains and their comparison is also well explained. By drawing attention towards an alarming cause, this piece of information will definitely help the physicians in that region to decide on the therapeutic strategies for CDI management.

Author Response

Response to Reviewer 1 Comments

POINT 1. Clostridioides difficile is a leading nosocomial agent that presents a major threat among healthcare regimens. Currently, the major mode of therapeutic management for Clostridioides difficile infection (CDI) is antibiotic therapy. However, evolution of antibiotic resistance is a serious, plaguing concern for the healthcare workers. In the current manuscript the authors have discussed this antibiotic resistance in C difficile strains Silesia region of Poland. The authors have done a very systematic study to assess the antibiotic susceptibility of these strains and their comparison is also well explained. By drawing attention towards an alarming cause, this piece of information will definitely help the physicians in that region to decide on the therapeutic strategies for CDI management.

RESPONSE to point 1:

Dear Reviewer 1

Thank you so much for the time you spent reviewing our article.

Also thank you for your positive opinion about our paper.

Sincerely

Prof. Dr Gayane Martirosian

Department of Medical Microbiology

Medical University of Silesia in Katowice, Poland

Reviewer 2 Report

Thank you for the opportunity to review this manuscript. Contributions to knowledge of CDI epidemiology and resistance are always of interest. This manuscript highlights the epidemiology in Poland which has a high prevalence of PCR ribotype 027 and 176 - this has previously been documented in the literature.

I think the results are potentially interesting but a little more interpretation is needed, particularly with regard to comparing with previous studies (eg. location, different breakpoints used, different epidemiology of ribotypes). At the moment,  some of this is not quite correct, and in some places not quite clear. Likewise, the introduction/discussion would benefit from a little attention to bring it up to date and make it a little more accurate.

I would be interested to know why E-testing was chosen instead of agar incorporation (gold standard) for the study.

Please take care not to confuse "strains" and "isolates". It is sometimes not clear what is meant.

Specific comments

 P2 line 43. What about other, more recent studies?

p2 line 44. Please include a reference for "other countries"

p2 line 45-47. Is this the correct reference? This paper describes clades, not ribotypes

p2 line 49. Please include a reference for this statement

p2 line 53. CDI can occur after treatment with antibiotics to which the organism is susceptible, likewise recurrence. Can the authors clarify what they mean by this statement please?

p2 line 59 replace "discarding repeated" with "removing duplicate"

p2 line 61-62. Was other antimicrobial use documented, or only FQ use?

P2 line 69. Why was the ribotype not determined for 9 isolates?

p6. line 117. "C. difficile ribotypes and resistance to antibiotics" does not seem to describe this particular section. I wonder if it would be more appropriate not to have a title fo rthis paragraph, and just have it as an introduction to the discussion? 

p6. line 137. I think, as this study only has 3 RT001 isolates and no RT012, while the study by Spigaglia et al examines 32 and 10 respectively, it is a little difficult to contrast the ermB results in these studies. It may be more useful to comment on clonal expansion of RTs containing ermB in particular locations, particularly as the outbreak of 001 in the US in the 1990s (Gerding et al 199) was linked to a clindamycin resistant, emrB + 001).

p6 lines 141-145. I think the emphasis on this section might need changing

In ermB-positive strains, GMs were significantly 143 higher than in ermB-negatives: 150.15 mg/L vs. 51.38 mg/L for erythromycin (p=0.0345) and 144 71.46 mg/L vs. 8.85 mg/L for clindamycin (p<0.0005), respectively: among 79 ermB-positive C. difficile isolates analyzed, in-vitro resistance to erythromycin 141 and clindamycin was found only in 62/79 (78.5%) strains belonging to RT 027 (n=60), RT 142 176 (n=1) and RT 001 (n=1).

Were the mechanisms for this explored any further? What might the reasons be for phenotypic resistance in these sioaltes without ermB?

line 167. The pan-European study data is a figure for all isolates across all countries tested, while the present study looks at a particular location with a high prevalence of a particular ribotype. The data for Poland from the pan European study - described in line 174, would be a more appropriate comparison and is comparable with the data described.

lines 176-190. What was the imipenem breakpoint in the study by Lachowicz et al? The imipenem breakpoint used in the study by Isidro et al was >16mg/L (also used in the Pan European study) however, in the present study the breakpoint is >4mg/L. If the breakpoints are different, then that will affect the interpretation of the data.

line200 - there have been publications in the previous year on metronidazole resistance mechanisms with roles for a plasmid, pCDMetro, and haem metabolism identified. For metronidazole in particular, the detection of resistance is affected by the medium used

line 211. In the 5 year pan European study by Freeman e al (published in 2020), there was a reduction in metronidazole resistant isolates over the course of the study.

line 222-224 - remove "whereas"

Lines227- 229 - the reference for this statement on van resistance and use in USA and China is a single centre study in China, which did not collet antimicrobial data. This statement should be revised or removed.

Lines 231-236

There are additional, more recent studies on MDR in C. difficile that should be added to add strength  (eg pan European study, AMR in CD analysis by Imwattana; Putsathit -JAC 2021)

Lines 277-290. Why was e-test selected for MIC determination? Gold standard method for C. difficile is agar incorporation. What medium was used for MIC determination?

For fluoroquinolones, there can be a range of MICs about 32mg/L - some isolates go up to 64 and 128mg/L. This should probably be made clear as it will affect GM MICs and make it difficult to compare between studies.

Author Response

Dear Editors and Reviewers

I would like to thank you very much for your review, and for your time spent reviewing our paper.  I prepare response for each of Reviewers comment, which I am sending now. 

Sincerely

Prof. Dr Gayane Martirosian

Reviewer 3 Report

Reviewer comments

Aptekorz et. al. have profiled antibiotic susceptibility of 215 isolates of Clostridioides difficile collected from feces of 215 patients (120 women & 95 men) across 13 hospitals in Silesia, Poland between December 2018-February 2019. They used C. diff specific GDH & Toxins A/B two enzyme immunoassays, and the C. diff strains were isolated on selective agar plates for further analysis. Isolates were tested for susceptibility to 11 antibiotics using E-test strips. Toxin (toxA, B, & CDT) and ermB  genes were screened from these isolates by multiplex PCR  and grouped for epidemic ribotypes (RT 027, 176, etc.). The authors concluded that C. dif incidence was higher, particularly the MDR RT 027/176 in the Silesia region. 

C. difficile is an important MDR pathogen associated with antibiotic use and causes CDI in people. Monitoring the incidence of this pathogen and their ribotype identification is useful in controlling the diseases. This article emphasizes this need. While the article is presented straight forward, there are some issues that need to be addressed.  

Major: 

The authors should discuss whether the jump in CDI cases in 2021 is likely due to the less access or care for these groups of CDI patients during the COVID-19 restrictions or how the pandemic impacted CDI reporting. This fact is critical in arriving conclusion. The discussion section should be shortened for each antibiotic.

The electrophoresis results of the mPCR can be included as supplementary data.

Minor:

- mention Clostridioides difficile infection (CDI) in the beginning 

- briefly describe in the introduction what is mean RT 027, 176, etc. means 

-  Statement in the abstract, “The incidence doubled from 26.4 in 2020 to 55.1 in 2021” can be moved to results or discussion and include citation(s). Also, see the COVID-19 related note above.

- line 80 and in other places; Please use consistent (MIC ug/ml instead of mg/L) wording throughout the manuscript that aligns with the data presented in table 2 (ug/ml).

-line 110, the result of e.g. the presence of .. not clear! delete e.g.

- please rephrase the sentence in lines 138-139, 195-196; and present short sentences, line 203.

-Reference: The taxonomical binomial system (C. difficile) needs to be followed.

Author Response

Below I attached responses to comments of Reviewer 3.

Sincerely

Prof. Dr Gayane Martirosian

Reviewer 4 Report

The article focused on the antibiotic resistance profile of RT 027/176 among Clostridioides difficile isolates in Silesia. There are some suggestions:

1.      In the first paragraph of Introduction. Line 27-39. Since this article focused on the resistance of C. difficile, suggest revise this paragraph to highlight the significance of antibiotics resistance.

2.      Line 36-37. “However, the underlying mechanisms remain elusive”. This sentence is vague. Please clarify this sentence.

3.      In Introduction part. Line 42-44. “The percentage MDR (defined as co-resistance….” Please cite reference for this definition.

4.      Line 42-44. Also please cite the reference for “….between 2.5 and 66% in various countries”

5.      Line 44-47 “Quite a few C. difficile ribotypes……” It is only an epidemiology data in Ref 5. It is very hard to have such conclusion.

6.      In Result part. Since the drug of choice for C. difficile is vancomycin and metronidazole, suggest to have more description for this two drugs.

7.      In Discussion part, suggest to move these paragraph into Introduction part instead of Discussion. Suggest to revise the whole Discussion part to focus on the major findings of this study.

Overall the finding is not important since most of these antibiotics are not drug of choice for CDI. The editing of Introduction and Discussion part should be revised.

Author Response

Response to Reviewer 2 Comments

Point 1:In the first paragraph of Introduction. Line 27-39. Since this article focused on the resistance of C. difficile, suggest revise this paragraph to highlight the significance of antibiotics resistance.

RESPONSE TO POINT 1: The first paragraph of Introduction, especially lines 37-39 were modified and additional sentence  “On the other hand,C. difficilewas recognized by CDC as one of the top 5 urgent antibiotic resistant threats in USA [5]” and reference no 5. were added (CDC Antibiotic Resistance Threats in the United States, 2019. CDC. Atlanta, GA: U.S. Department of Health and Human Services2019, doi:10.15620/cdc:82532.).

Point 2:  Line 36-37. “However, the underlying mechanisms remain elusive”. This sentence is vague. Please clarify this sentence.

RESPONSE TO POINT 2: The sentence  “However, the underlying mechanisms remain elusive” was removed, because this paper is focused to discuss antibiotic resistance of C. difficileisolates. In the first paragraph of Introduction only a general information about C. difficilepathogenicity is included.

 Point 3:In Introduction part. Line 42-44. “The percentage MDR (defined as co-resistance….” Please cite reference for this definition. 

RESPONSE TO POINT 3: From lines 42-44 the definition of MDR (defined as resistance to moxifloxacin, clindamycin, erythromycin, imipenem and rifampicin) was removed, because MDR usually is defined as a resistance to 3 or more classes of antibiotics. In various articles, authors also select additional antibiotic classes to define MDR, so, we decided to remove this definition and present it later in Results section.

Point 4:Line 42-44. Also please cite the reference for “….between 2.5 and 66% in various countries”

RESPONSE TO POINT 4: To lines 42-44 (now 48-51) references no. 6 and 7 were added: no 6 (Spigaglia, P.; Barbanti, F.; Mastrantonio, P.; European Study Group on Clostridium difficile (ESGCD) Multidrug Resistance in European Clostridium Difficile Clinical Isolates. Journal of Antimicrobial Chemotherapy2011,66, 2227–2234, doi:10.1093/jac/dkr292.) and no 7 (Barbanti, F.; Spigaglia, P. Microbiological Characteristics of Human and Animal Isolates of Clostridioides Difficile in Italy: Results of the Istituto Superiore Di Sanità in the Years 2006–2016. Anaerobe2020,61, 102136, doi:10.1016/J.ANAEROBE.2019.102136).

And we added a sentence: ”European prospective study of CDI indicated that 55% of resistant clinical isolates were MDR in 2005 [6]. MDR C. difficile strains resistant to rifampicin were described in Italy [7] and other countries.”

Point 5: Line 44-47 “Quite a few C. difficile ribotypes……” It is only an epidemiology data in Ref 5. It is very hard to have such conclusion.

RESPONSE TO POINT 5: According to Reviewer comment lines 44-47 (now 43-47) were modified  as “Some epidemiological data suggest that quite a fewC. difficileribotypes have been associated with specific antibiotic resistance, including strains resistant to fluoroquinolones - RT 027 and RT 017, rifampicin - RT 027, and clindamycin - RT 017 or tetracycline - RT 078 [8]” 

Point 6: In Result part. Since the drug of choice for C. difficile is vancomycin and metronidazole, suggest to have more description for this two drugs.

RESPONSE TO POINT 6: According to Reviewer suggestion GM values for metronidazole and vancomycin – lines 92-96 (drugs of choice for treatment of CDIs at times of C. difficilestrains collection for this study) were added to Results section from Table 2 “The GMs (geometric means) of metronidazole for all tested ribotypes was 0.68 mg/L (RT 027 – 1.0 mg/L, RT 176 – 1.1 mg/L, other toxigenic strains – 0.13 mg/L, non-toxigenic strains – 0.33 mg/L). The GM of vancomycin for all tested ribotypes was 0.25 mg/L (RT 027 – 0.26 mg/L, RT 176 – 0.16 mg/L, other toxigenic strains – 0.22 mg/L, non-toxigenic strains – 0.36 mg/L) (Table 2).

Since we saw the problem in Table 2, part of which was invisible, we had to make changes to this Table: Geometric Means was changed to abbreviation GM and Strains Resistant – to SR (changes are included in Legend part of Table 2).

Point 7:  In Discussion part, suggest to move these paragraph into Introduction part instead of Discussion. Suggest to revise the whole Discussion part to focus on the major findings of this study.

RESPONSE TO POINT 7: We revised the Discussion section, according to Reviewer suggestion, and sentence from lines 99-105 was moved to Introduction section: “The global spread and outbreaks of hyperepidemic C. difficilestrains belonging to RT 027 is associated with antibiotic usage, especially with the massive use of fluoroquinolones. However, there are also outbreaks of CDI in both American and European healthcare facilities, caused by strains belonging to other ribotypes, such as 001, 002 and 014/020, as well as 017, 018, 106, 176 and 244 [2,9-11]. Exposure to antimicrobials plays a significant role in the pathogenesis of CDI, and resistance should be considered a predisposing factor for CDI development.”– now lines 55-61.

Additional point without number: Overall the finding is not important since most of these antibiotics are not drug of choice for CDI. The editing of Introduction and Discussion part should be revised.

RESPONSE:In Discussion section we payed attention on our findings described in Results and also presented additional data from medical literature concerned to other mechanisms of C. difficileresistance.

In our opinion, mechanisms of resistance, described in this paper are important from epidemiological point of view, especially those described in hypervirulent RT 027 C. difficilestrains – dominant in our region.

We would like to say to Reviewer 2 thank you very much for the time spent reviewing our article. Thank you once again for such a detailed analysis of this paper. We hope that our responses will satisfy the Reviewer and the article can be accepted for publication in Pathogens.

Sincerely

Prof. Dr Gayane Martirosian

Department of Medical Microbiology

Medical University of Silesia in Katowice, Poland

Round 2

Reviewer 4 Report

had been revised as suggestion

Author Response

Dear Reviewers 4,5

Thank you very much for your time spent reviewing our paper and for your positive comments.

Sincerely

Prof. Dr Gayane Martirosian 
